# A Bloody Conspiracy— Blood Vessels and Immune Cells in the Tumor Microenvironment

**DOI:** 10.3390/cancers14194581

**Published:** 2022-09-21

**Authors:** Lisa Terrassoux, Hugo Claux, Salimata Bacari, Samuel Meignan, Alessandro Furlan

**Affiliations:** 1Univ. Lille, CNRS, Inserm, CHU Lille, UMR9020-U1277-CANTHER-Cancer Heterogeneity Plasticity and Resistance to Therapies, F-59000 Lille, France; 2Tumorigenesis and Resistance to Treatment Unit, Centre Oscar Lambret, F-59000 Lille, France

**Keywords:** tumor, angiogenesis, immunity, infiltration, paracrine, microenvironment, chips, microfluidics, organoids

## Abstract

**Simple Summary:**

The tumor microenvironment has risen over the last years as a significant contributor to the failure of antitumoral strategies due to its numerous pro-tumorigenic activities. In this review, we focused on two features of this microenvironment, namely angiogenesis and immunity, which have been the targets of therapies to tackle tumors via its microenvironmental part over the last decade. Increasing our knowledge of the complex interactions within this ecosystem is mandatory to optimize these therapeutic approaches. The development of innovative experimental models is of great help in reaching this goal.

**Abstract:**

Cancer progression occurs in concomitance with a profound remodeling of the cellular microenvironment. Far from being a mere passive event, the re-orchestration of interactions between the various cell types surrounding tumors highly contributes to the progression of the latter. Tumors notably recruit and stimulate the sprouting of new blood vessels through a process called neo-angiogenesis. Beyond helping the tumor cope with an increased metabolic demand associated with rapid growth, this also controls the metastatic dissemination of cancer cells and the infiltration of immune cells in the tumor microenvironment. To decipher this critical interplay for the clinical progression of tumors, the research community has developed several valuable models in the last decades. This review offers an overview of the various instrumental solutions currently available, including microfluidic chips, co-culture models, and the recent rise of organoids. We highlight the advantages of each technique and the specific questions they can address to better understand the tumor immuno-angiogenic ecosystem. Finally, we discuss this development field’s fundamental and applied perspectives.

## 1. The Tumor Microenvironment: The Octopus Spreading Its Tentacles

### 1.1. The Tumor Vascularization, a Key Partner in the Malignant Progression

Cancers represent a major health issue, accounting for nearly 10 million deaths in 2020, according to the World Health Organization [1]. Concomitantly with the identification of the intrinsic genetic drivers of tumorigenesis, the research community has evidenced over the last decades the critical importance of the tumor microenvironment in the evolution of the pathology.

Tumors progress in concert with their neighbors and orchestrate a kind of conspiracy to promote their malignant development [2]. Among these “partners in crime”, the vasculature plays a key role, notably by providing the tumors with nutrients and oxygen required for their sustained growth, and an angiogenic switch is typical during malignant progression. Almost half a century ago, Judah Folkman introduced this concept of tumor angiogenesis, namely the fact that solid tumors induce the development of neo-vessels for their profit [3]. Both tumor and vascular cells cooperate in this ecosystem to foster the growth of each other. Since then, many studies have investigated and elucidated the complex interactions between tumors and blood vessels, leading to the elaboration of anti-angiogenic therapeutic strategies in clinics [4,5,6]. Unfortunately, so far, the success of such strategies has been mitigated due to the high plasticity of cancer cells and the heterogeneity and variability of abnormal tumor vessels that allow this ecosystem to bypass the attempts to block it [7].

Blood vessels physiologically consist of an inner layer of epithelial cells called endothelial cells, laid on a basal lamina. Perivascular cells encompass smooth muscle cells, which are essential for arterial contractility, and pericytes in the case of capillaries, the small blood vessels responsible for most exchanges with tissues. Tumor vascular networks are highly anarchic, with uncontrolled sprouting leading to excessive branching and blind ends. Tumor-associated endothelial cells display abnormal morphology, resulting in disorganized monolayers and imperfect coverage by pericytes. Tumoral blood vessels are, therefore, often functionally defective in terms of perfusion but also permeability, suggesting that one of their critical roles would reside in the facilitation of cancer cell dissemination in the systemic circulation through the metastatic process, beyond the trophic supply that they provide [8]. In contrast with these permeable vessels, when it comes to thinking about brain tumors, one also has to take into account the very particular organization of the brain capillary endothelial cells which, together with pericytes and astrocytes, contribute to the impermeability of the Blood–Brain Barrier (BBB) [9]. Consequently, the BBB impairs most compounds from diffusing from the luminal compartment of blood vessels towards the brain parenchyma and brain tumors. We will discuss this point later in this review since brain tumors in this particular context require specific study models.

### 1.2. The Cocktail Driving Tumor Angiogenesis

Tumor neo-angiogenesis relies on a cocktail of soluble factors secreted in the tumor microenvironment: cytokines, growth factors, extracellular matrix (ECM) proteases, etc. [10]. They can be produced both by cancer cells and stromal cells, including fibroblasts, immune cells, and endothelial cells as well, leading to reciprocal dialogs between the different partners modulating both angiogenesis and immune responses.

Among the first identified molecules within this pro-angiogenic array, the VEGF (Vascular Endothelial Growth Factor) family has been the topic of interest in many studies. VEGF-A binding on its receptors at the cell membrane of endothelial tip cells will guide vessel sprouting, by promoting the local degradation of the extracellular matrix required for cells to invade it. Gradients of VEGF-A thus induce the motility of these tip cells, while additional molecules such as delta ligand-like 4 (DLL4) will inhibit the sprouting of the following stalk cells, with the concomitant orchestration of ECM deposition and remodeling [11].

One of the main biophysical drivers of neo-angiogenesis in tumors resides in the decrease in tissue oxygen partial pressure, ultimately leading to the secretion of pro-angiogenic factors. Hypoxia leads to a loss of HIF1α (Hypoxia-Inducible Factor) hydroxylation, which induces its stabilization (when hydroxylated, under normoxia, HIF1α is degraded by the VHL (Von Hippel Lindau) Ubiquitin ligase). This will trigger a transcriptional program allowing the expression of genes aimed at counterbalancing hypoxia. These include pro-angiogenic factors, such as VEGF-A, CXCL12 aka SDF-1 (Stromal Derived Factor), ANGPT2 (Angiopoietin 2), and PDGF-B (Platelet-Derived Growth Factor) notably [12] (Figure 1A). In that manner, the subsequent formation of neo-vessels should improve the oxygen supply and restore a new equilibrium.

Positive and negative feedbacks occur within this dialog, which renders the results of anti-angiogenic therapies upon tumor progression more complex to decipher than initially thought. For example, VEGF can repress Epithelial-to-Mesenchymal Transition and invasion of cancer cells via forming a MET/VEGFR2 complex [13]. In the same vein, antiangiogenic therapy was shown to promote malignant tumor progression in some contexts [14]. Nevertheless, tumor invasion and angiogenesis also share common molecular mechanisms that make them two birds of a feather. For example, HGF/SF (Hepatocyte Growth Factor/Scatter Factor) favors both cancer cell and endothelial cell invasion (for review, see [15]). This also applies to extracellular matrix proteases, notably the MMPs that can be secreted by tumor cells and recruited at the plasma membrane of endothelial cells to favor angiogenesis in a paracrine manner [16]. Interestingly, immune cells also express MMPs and contribute to creating a pro-invasive local environment [10], which can be efficiently targeted to impair tumor progression in cervical cancer models [17]. More globally, it was shown that several soluble factors secreted when cancer cells undergo Epithelial-to-Mesenchymal Transition, including InterLeukins IL-6 and IL-8, and GM-CSF (Granulocyte-Macrophage Colony Stimulating Factor), stimulate angiogenesis and the recruitment of myeloid cells [18].

### 1.3. The Recruitment of Undercover Agents

Altogether, the aforementioned factors classically lead to a pro-inflammatory environment in which the migration of cancer cells is increased, together with that of angiogenic endothelial cells and immune cells. Let us focus now on the recruitment and the impact of these latter in the tumor microenvironment.

Tumors recruit many cell types present in their neighborhood to help their progression, and immune cells definitely fall into this category [19] (Figure 1B). Once recruited, immune cells can play various roles, and the global immune landscape in tumors evolves along their progression [20]. While their primary purpose is to suppress tumor growth, immune cells may become beneficial to the tumor rather than detrimental. In that frame, a recent study evidenced that targeting macrophages could improve the survival of glioma-bearing mice [21]. It has to be noted that inflammation can even precede and promote tumorigenesis, as exemplified by *Helicobacter pylori*-driven gastritis or inflammatory bowel diseases [22].

Each immune cell type will express some chemokine receptors to respond to particular signals. Macrophages represent the first type of immune cells frequently detected in the tumor microenvironment and classically derive from circulating monocytes attracted by chemokines such as CCL2, CXCL5, and CXCL12 secreted by tumors [23]. CXCL12 notably contributes to the recruitment of Tumor-Associated Macrophages (TAM) in hypoxic regions, as evidenced in murine astrocytoma models [24]. Effector T lymphocytes and NK (Natural Killer) cells express CXCR3 and migrate into tumors in response to CXCL9 and 10, while immunosuppressive T reg lymphocytes express CCR4 and CCR10 that make them sensitive to the presence of CCL22 and CCL28. Interestingly, hypoxia induces the expression of CCL28 within tumors, which contributes to cancer immune tolerance [25]. Though less under the spotlight, eosinophils are also recruited in the tumor microenvironment, preferentially in hypoxic zones, thanks to CCL11 [26]. Concerning neutrophils, chemotaxis is ensured by CXCR1 and CXCR2 agonists, which favor at the same time the release of neutrophil extracellular traps (NETs) that can wrap tumor cells and protect them from T cells and NK cells [27]. These examples (recapitulated in Table 1) are only a few among the plethora of cytokine signaling mechanisms leading to immune cell recruitment within tumors.

While tumors favor the recruitment from the blood circulation of immune cells, it is also interesting to observe that immune cells reciprocally favor tumor angiogenesis, thus engendering a vicious circle. To do that, the various immune cells release pro-angiogenic factors. Mast cells, for example, secrete VEGF-A, FGF2 (Fibroblast Growth Factor), CXCL8, and MMPs [28]. The production of VEGF by myeloid cells also contributes to tumor angiogenesis, as illustrated by the delay in the angiogenic switch in a mouse tumor model when the *vegfa* gene is conditionally depleted in the myeloid lineage [29]. Moreover, infiltrating cell-derived MMP9 contributes to such a tumor angiogenic switch, possibly through the MMP-mediated release of VEGF trapped in the extracellular matrix (ECM) [30]. Finally, immunosuppressive T reg lymphocytes also promote angiogenesis [25], although they globally tend to decrease the pro-inflammatory responses around tumors [31].

**Table 1 cancers-14-04581-t001:** An overview of cytokines in the immuno-vascular tumor microenvironment.

Cytokines	Effects	References
IL-6/IL-8	Angiogenesis promotion/inflammation	[18]
GM-CSF	Angiogenesis promotion	[18]
CCL2	Monocyte recruitment	[23]
CXCL5	Monocyte recruitment	[23]
CXCL12 (also called SDF1)	TAM recruitment/	[23,24]
CXCL12 (also called SDF1)	Dendritic cell migration	[32]
CXCL9/CXCL10	TL and NK recruitment	[23]
CCL22/CCL28	T-reg recruitment	[25]
IL-1/TNFα	Inflammation/Haptotaxis	[33]
CCL11	Eosinophil recruitment	[26]
CXCL1/CXCL2/CXCL8	Neutrophil recruitment	[27]

Many other cell types in the tumor microenvironment also play instrumental roles in regulating tumor progression. Notably, Cancer-Associated Fibroblasts (CAF) produce and remodel the extracellular matrix in the vicinity of tumors, and establish critical signaling interactions with tumor cells, which will promote their growth and resistance to treatments [34], but they also dialog with immune cells infiltrating the tumors [35] and influence angiogenesis [36].

### 1.4. The Underground Environment: Of Hypoxia and Its Malignant Consequences

As illustrated hereinabove, hypoxia plays a crucial role in malignant progression by triggering the secretion of pro-angiogenic and chemoattractant molecules. Still, this is not the only mechanism through which hypoxia will deeply impact the tumor microenvironment. Indeed, even if it is long known that hypoxia rewires metabolic signaling in cancer cells, recent studies put it back in a more extensive context including cancer cell neighbors. Exacerbated glycolysis in hypoxic zones leads tumor cells to secrete high amounts of lactate in the tumor parenchyma. Far from being a mere final by-product, lactate can influence the cellular physiology in these regions. A symbiosis was evidenced between hypoxic cancer cells and normoxic cells in the vicinity that can internalize and metabolize lactate [37]. Endothelial cells can also import this lactate through the monocarboxylate transporter 1 (MCT1), and this process was shown to promote angiogenesis [38]. Interestingly, endothelial cells from tumor vessels display increased glycolysis, which was efficiently used to normalize these vessels and reduce metastasis in a melanoma xenograft model [39]. Other strategies aimed at impairing angiogenesis and metastasis by tackling the endothelial cell metabolism with a cholesterol anabolism enzyme inhibitor also displayed promising results [40].

To come back to hypoxia and lactate, tumor-derived lactate also impacts immune cells, as demonstrated by its induction of Tumor-Associated Macrophage polarization in an M2-like state [41]. Moreover, glycolysis-driven lactate secretion is most often associated with an acidification of the extracellular compartment, which was shown to disrupt extracellular vesicles containing pro-angiogenic molecules such as VEGF-A and MMPs [42]. The global impact of lactate and acidic pH on the modulation of immune functions is reviewed in more detail in [43]. Since immune cells within the tumors must face harsh nutritional conditions, it is also essential for them to adapt to this metabolic stress, and autophagy can be activated both in the tumoral and immune cells and is involved in tumor immune control [44].

### 1.5. The Remote Expansion of Activities

Most cancer-associated deaths are due to metastasis, namely the growth of secondary tumors at distance. This phenomenon occurs through a multi-step process called the metastatic cascade (Figure 1C), initiating with intravasation of cancer cells, their survival in the systemic circulation, their extravasation at distant organs, and their final growth from cryptic to overt metastatic foci. Interactions of cancer cells with endothelial cells are obviously crucial during the transendothelial migration steps, and the creation of a pro-invasive inflammatory microenvironment represents a facilitating parameter as well. Consistently, angiogenesis and inflammation are associated with metastasis in various tumors, notably breast cancers [45].

Among immune cells, macrophages are among the first to be recruited in primary tumor sites. Breast cancer mice models allowed it to be demonstrated that paracrine loops established between tumoral cells and macrophages led to a synergistic EGF (Epidermal Growth Factor) and CSF1 (Colony Stimulating Factor) mediated stimulation of the invasive potential of both cell types [46]. Moreover, multiphoton microscopy demonstrated that perivascular macrophages assisted tumor cell intravasation in these models [47].

Globally, various immune cell types are involved in the different steps of the metastatic cascade, by protecting cancer cells and/or by favoring their transmigration [48]. Besides the previously mentioned case of neutrophils forming a shield around tumor cells [27], macrophages can be tethered to breast cancer cells via VCAM-1 and favor their survival during the development of lung metastases [49].

In addition, tumors also send signals that generate premetastatic niches, i.e., distant regions in which VEGF-R1 positive bone marrow-derived cells cluster before the arrival of metastatic cancer cells [50]. Such places display immuno-suppressive signatures driven by myeloid cells, and immunomodulatory strategies were able to efficiently reduce metastatic growth in these settings [51]. In melanoma as well, tumor cells can reprogram bone marrow progenitors and prepare pre-metastatic sites through the release of exosomes [52]. Strikingly, immune cells also use exosomes for remote immune surveillance tasks, as illustrated by the cytotoxic potential of NK cell-derived exosomes [53].

Taken together, these data highlight the intricate links between tumoral and immune cells along the different steps of metastasis, and the potential of immune therapies to fight this dismal trait of tumors [54].

Clinical trials aimed at fighting cancers via anti-angiogenic and anti-immune approaches were developed over the last decades, and combining these strategies provided encouraging results in recent years (Figure 2).

## 2. Chips, Patrolling to Catch Specific Scenes of the Bloody Conspiracy

After this overview of the tumor immuno-angiogenic phenotype, let us focus on the ex vivo tools developed by the research community to help unravel its different aspects. Historical models based on the transplantation of sections of animal blood vessels, such as the aortic ring assay, and more recent microvascular networks from rat mesentery, proved useful to study angiogenesis [55]. Still, their use tends to be restrained due to ethical issues associated with animal experimentation and alternative ex vivo models were developed, which greatly facilitated the real-time visualization of cellular events.

### 2.1. Tumor Angiogenesis under the Spotlight Thanks to VoC, Vessels-on-Chip

In this chapter, we will describe the different categories of chips implemented to mimic tumor vessels and their neighbors. Herein, we will use the term “chip” to refer to a microfluidic device with a pattern of microchannels that process tiny amounts of cells and liquids. Vessels-on-Chip (VoC) can display different configurations (single channels, parallelized channels, or networks) depending on the question researchers want to address. Their constitution is based on different methods that have evolved over the last few years.

The oldest and most straightforward method, in principle, to create a channel within an extracellular matrix gel was to insert a needle into these materials and then remove it (Figure 3A). However, the removal of the needle might not always be straightforward and often resulted in an alteration of the overall structure. In addition, this method was limited to forming a single channel, narrowing its potential applications. Using polymers such as silicone or PDMS (PolyDiMethylSiloxane) to fill molds designed on purpose allowed the generation of a wider variety of geometries, depending on the vascular aspect under investigation. Yet, such models required coating these synthetic polymers with appropriate extracellular matrix components [56], adding to the protocol a tricky step that was frequently troublesome, with the final obtention of inhomogeneous coating and subsequently ill-controlled discontinuous vessels.

To bypass this issue, an original method was proposed and developed: the VFP (Viscous Finger Patterning), which relies on the principle that within a lumen, a less viscous fluid displaces a more viscous fluid. Thanks to the passive pumping of cell culture medium through an ECM hydrogel under polymerization, researchers succeeded in generating a continuous ECM layer within channels of different shapes (straight, curved, Y-shaped) [33,57]. This method was recently adapted to create endothelial vessels covered by perivascular cells with each cell type sequentially seeded by VFP [58].

These VoCs are very helpful in deciphering the molecular determinants of vessel physiology. When an ECM layer surrounds the channel, the VoC can, for example, be used to investigate vessel sprouting thanks to pharmacological and/or genetic approaches [59]. Of note, one can also use VoCs to quantify vascular permeability by monitoring over time the diffusion across the vascular wall of compounds coupled with fluorophores [60]. In this context, the presence of perivascular cells decreases the permeability of the VoC by increasing the tightness of the endothelial barrier [58].

Implementing microfluidics within VoCs permits recapitulating molecular gradients and showed that VEGF and angiopoietin 1 gradients orchestrate the migration of tip and stalk cells during sprouting angiogenesis [59]. Moreover, the integration of tumor cells within the VoC can drive sprouting and allows the study more specifically of tumor angiogenesis, as illustrated by the co-culture of ccRCC (clear cell Renal Cell Carcinoma) clusters around HUVEC channels [60]. In parallel, microvascular branched networks were also generated within tissue chambers and co-cultured with tumor cell clusters, to investigate the response to therapies of tumor clusters and/or their vasculature [61]. More recently, a chip was developed to generate a hypoxic tumor and observe how it can induce angiogenic sprouting from an adjacent endothelial channel [62] (Figure 3C).

### 2.2. A Successful Business Requires a Steady Cash Flow

In endogenous vessels, the shear stress induced by blood flow represents a key regulator of blood vessel physiology, and shear stress notably drives the reorientation and reinforcement of cytoskeletal fibers within endothelial cells [61]. The integration of microfluidics into chips allows recapitulation of this critical process in experimental models (Figure 3B), and researchers could thus demonstrate that shear stress dampened vascular sprouting via nitric oxide signaling [62]. Moreover, in the same study, thanks to the use of a second channel to generate VEGF gradients, the authors evidenced that positive and negative VEGF gradients respectively triggered sprouting or dilation of vessels. Interestingly, another study completed these works and showed that flow stagnation at bifurcation branching points blocked the sprouting as well, while a transvascular flow, controlled by modulating interstitial fluid pressure in an adjacent column, favored, in contrast, the vascular sprouting [63]. Besides this shear stress issue, the blood flow is responsible for the delivery of drugs to tumors, and VoCs can be used to follow the growth and intravasation of tumor cells in response to treatments [64]. Strikingly, human gut, liver, and kidney-on-chips developed around endothelial channels recently allowed recapitulation of the pharmacokinetics (PK) of cisplatin in patients, thus opening unique perspectives for the pre-clinical assessment of antitumoral drugs [65].

### 2.3. What about Particular Supply Routes?

Let us now focus on two particular physiological conditions faced by tumor vessels, in the lung and the brain. The lung capillaries constitute a network tightly associated with epithelial alveoli. Recently, a smart design aimed at mimicking the breath of epithelial alveoli coupled with an endothelial channel allowed recapitulation of lung cancer growth and evidenced the impact of breathing on cancer cell intravasation and response to treatments [66]. This paves the way for a better ex vivo assessment of treatments to target lung tumors. Concerning brain tumors, they interact with the highly impermeable vessels of the Blood–Brain Barrier. The canonical models to recreate the BBB are based on the co-culture of cells upon inserts containing microporated membranes. Over the years, the choice of cells that can efficiently trigger the formation of tight junctions by endothelial cells and ensure proper filtration of compounds as expected with the BBB has been improved and adapted for the specific study of tumors [67]. Ultimately, BBB models constituted exclusively of human cells were used to investigate interactions between the BBB and pediatric glioma cells [68]. The development of PDMS-based microfluidic chips led to the development of additional models, and recently a study, based on the rationale that the embryogenic BBB during its genesis faces hypoxia, showed that hypoxia could improve the functionality of such BBB models [69]. The same research group also proposed a BBB channel implemented thanks to the VFP process that allowed the study of the importance of pericytes and astrocytes for the permeability of the endothelial channel [70], which could be very useful in studying the interactions between the BBB and brain tumor cells. Finally, 3D bioprinting was also implemented to set up geometries of tumor-BBB interactions within chips [70,71], although quantifying the proper barrier function in such models has been quite tricky so far.

### 2.4. The Choreography of Immune Cells as Undercover Agents among Gangsters

Chips also constitute precious tools for understanding the complexity of the cancer immune environment and how it assembles.

Before being able to take action on-site, immune cells need to reach and infiltrate the said site. For that purpose, circulating immune cells need to recognize where they should exit the bloodstream to reach the nearby tissue. Haptotaxis, i.e., the guidance via a gradient of adhesive cues, constitutes the first step in this process. The adhesion of immune cells to the endothelial surface can be quantified in VoC, and the use of pro-inflammatory cytokines such as TNFα or IL-1 increases this adhesion, in correlation with the overexpression of ICAM-1 or VCAM-1 adhesion molecules [58]. In addition, thanks to a microfluidic system under flow, Arts et al. shed light on the existence of “hotspots” with enrichment in PECAM-1 and ICAM-1 adhesive molecules at junctional membrane protrusions between endothelial cells, where neutrophils preferentially undergo diapedesis after crawling along the junction up to these specific sites [72]. Moreover, a study based on an inflammation-on-a-chip model revealed that the ECM composition modulated the capacity of neutrophils to undergo diapedesis in response to TNFα, with an exacerbated transmigration occurring in vessels grown in basement membrane extract gels [73].

Physiologically, immune cells sense and respond to gradients of chemokines, according to so-called chemotaxis (see Section 1.3), which can be efficiently reproduced and investigated in chips. An exciting system to study immune chemotaxis was developed by Um et al. in the form of microfluidic mazes, highlighting the increased chemotactic potential of cancer cells, when compared to normal cells, towards immature dendritic cells [74]. To step further into mimicking in vivo conditions, the idea of adding a “barrier” between the tumoral and immune channels to represent the tissular and circular compartments, respectively, was implemented. By adding tiny connecting channels that limit the exchanges between compartments, Parlato et al. were able to understand that the migration process of dendritic cells within a tumor environment was guided by the CXCR4/CXCL12 couple [32]. The recruitment of neutrophils towards tumors could be recapitulated thanks to another device, with a microfluidic flow over a porous membrane located upon a collagen gel encompassing 3D tumor clusters [75]. Interestingly, the authors demonstrated that while they were being recruited by ovarian cancer cells, neutrophils promoted the collective invasion of cancer cells within the surrounding ECM, notably by releasing neutrophil extracellular traps (NETs). Using a chip with a central tumor compartment surrounded by a vascular channel and an immune compartment allowed Cui et al. to investigate the interplay between glioblastoma (GBM) and immune cells. They evidenced that mesenchymal subtypes attracted many Tumor-Associated Macrophages (TAM), but few cytotoxic T-cells [76]. This system also demonstrated that GBM cells triggered an M2-like polarization in TAM, contributing to immunosuppression and angiogenesis [77].

Chips also allow studying immune-mediated antitumoral cytotoxicity. Hence, a microfluidic device was generated to recapitulate a hypoxic 3D tumor surrounded by channels through which CAR (Chimeric Antigen Receptor) T cells are delivered [78] (Figure 3C). Thanks to this model, Ando et al. were able to evaluate the infiltration of CAR-T cells and their propensity to induce the cell death of ovarian cancer cells as a function of hypoxia generated within the chip. Although not investigated in this study, it would be interesting to study the hypoxia-driven expression of chemokines that could attract CAR-T cells and other immune cells in this model (see Section 1.3). In the same vein, another tumor-on-chip based on the depletion of nutrients along an axis perpendicular to the feeding channel highlighted that such a depletion induced by nutrient consumption by tumor cells led to an exhaustion of NK cells that persisted over time [79]. Interestingly, administering checkpoint inhibitors could dampen this exhaustion, which illustrates one of the many beneficial facets of such drugs. Such immunocompetent cancer-on-chip models are therefore much helpful in evaluating the potential of antitumoral immune-based therapies [80].

### 2.5. Crossing the Legal Line

Transendothelial migration is not exclusively done by immune cells since cancer cells also cross the endothelial barrier during the early and late phases of the metastatic process, via intravasation and extravasation mechanisms, respectively. Microfluidic chips also offer interesting designs to control fluid pressures within or between compartments and were efficiently used to highlight the impact of luminal and trans-endothelial flows upon the extravasation of cancer cells from microvascular networks [81]. They nicely evidenced that a luminal flow in the same range as that of blood flow within capillaries favored the extravasation of breast cancer cells. Moreover, by applying differential pressures, the authors could recreate intramural flow to mimic the interstitial fluid pressure and demonstrated that it increased the tumor cell transmigration speed and their subsequent migration in the perivascular matrix.

Another biophysical parameter of the tumor microenvironment that affects the extravasation process is hypoxia, which can favor the extravasation of cancer cells through a VoC in a HIF-1α-dependent manner [82]. In the framework of the interplay between cancer, endothelial, and immune cells, the same research group also demonstrated that the presence of human neutrophils within the vascular networks significantly enhanced the proportion of extravasated cancer cells [83], and reported that neutrophils were entrapped by tumor cells via chemotactic confinement driven by autocrine and paracrine mechanisms [84]. In the same vein, but in distinct devices based on two adjacent channels, macrophages were shown to favor breast cancer cell invasion [85] and intravasation [86], thus highlighting how important immune cells are in the regulation of cancer cell transendothelial migration. Vertical systems with microfluidic channels and porous membranes were also developed to study the intravasation process and allowed one to visualize that different cancer cell lines can use either transcellular or paracellular paths to cross the endothelium (for review, see [87]).

## 3. Future Is Coming, Giving a New Dimension to the Trade

### 3.1. Spying on the Double Game of Endothelial Cells within Tumors

Blood vessels can be mimicked in chips, with appropriate luminal and abluminal compartments, yielding a precious amount of knowledge on vessel biology. Yet, these tools remain basic representations of the original 3D vessel geometry, which is usually much less straight. Besides the aforementioned chips, complementary development tracks were therefore pioneered over the last years to obtain further insight into the 3D interplay between tumoral, endothelial, and immune cells. In that frame, 3D gel-based models have emerged, especially with the rise of organoids [88]. Such models better recapitulate the tissular geometry and proper cellular interactions, thus ensuring a better reproduction of endogenous physiology [89]. In oncology, 3D models proved to be efficient for high-throughput drug screening in the context of personalized medicine [90,91]. That’s why, although they classically represent more constraining models from an experimental point of view, many efforts are made nowadays to study cancer processes in 3D ECM gels. This is especially relevant when one wants to investigate heterotypic interactions and study migration and chemotaxis events. Indeed, co-culture models in 3D hydrogels allow mimicking heterotypic dialogs occurring between different cell types that can be spatially precisely organized, especially now that bioprinting has become widespread.

It has to be noted that the mechanical properties of ECM components deeply influence the phenotype of cells cultured within and should be taken into account when designing the experiments and choosing the most relevant scaffold [92]. Biomaterials used in the field of oncology display a wide variety of properties [93]. Collagens are a major component of ECM and can represent as much as 30% of the total protein mass in mammals. Fibrillar collagens such as type I collagen, widely used in bioassays, allow tensile strength to be exerted by cells, and can be easily remodeled by tumor cells. In addition, collagen concentrations can be varied to enable a range of ECM stiffness with elastic moduli from 150 to 1200 Pa, which was used to mimic matrix stiffening classically associated with tumor progression and to highlight the change in signaling and phenotype of mammary cells as a function of tissue stiffness [94]. In the field of epithelial organoids, the gold standard ECM is reconstituted basement membrane, mainly consisting of laminin and type IV collagen, which controls structure polarization. Hyaluronic acid is another highly abundant matrix, especially important to reproduce a relevant microenvironment for glioblastoma cells since the brain and its tumoral counterpart are rich in this component. It retains water and, when properly cross-linked, is amenable to deformations, which represents an interesting asset for manipulating hydrogels without destroying them. Finally, though not present in animal tissues, alginate displays several assets for cancer cell 3D culture, including its fast gelation and its thermal stability, and can allow encapsulated cells to produce their own matrix.

The first in vitro 3D models for blood vessels used suspensions of endothelial cells seeded upon, or embedded within, type I collagen or basement membrane extracts. In these conditions, endothelial cells self-organized to give rise to rudimentary capillary-like structures (Figure 4A). Interestingly, the incorporation of other cell types within these models highly improved the complexity of the vascular structures that formed, highlighting how endothelial cells are tightly dependent on their neighbors to express their complete phenotype. For example, HUVEC cells grown in fibrin gels require the presence of fibroblasts above the gels to develop 3D sprouting vessels [95]. The impact of tumor cells upon endothelial vessels is obviously important, and the development of tumoroids including a vascular network will be helpful to understand this network’s plasticity and its response to anti-angiogenic drugs [96].

Moreover, 3D models have allowed highlighting reciprocal interactions between cancer cells and endothelial cells [97,98]. Interestingly, these in vitro 3D models evidenced that endothelial cells could exert an angiocrine influence upon cancer cells without any metabolic supply. This can be paralleled with the observations of endothelial cells emitting inductive signals to their neighbors during embryogenesis, before forming functional blood vessels [99,100]. Of note, this further widened the impact of blood vessels in the tumor ecosystem and was also observed in in vivo models. Thus, a recent study with a zebrafish xenograft model highlighted that tumor growth was stimulated by the presence of endothelial cords before any vascular function [101]. In the same vein, it was shown that endothelial cells present in the tumor microenvironment could alter the response to therapies, beyond their function of blood supply [102].

### 3.2. Unmasking the Interplay between Tumoral Thugs and Undercover Immune Agents

Such 3D models are also useful for deciphering the dialog between immune and tumor cells occurring via paracrine interactions [102]. Strikingly, the 3D context is mandatory for the expression of some phenotypic traits and notably favors the expression by cancer cells of pro-inflammatory and pro-angiogenic cytokines, such as IL-8, which promotes the invasion of endothelial cells [103]. Interestingly, organoids derived from liver carcinoma and co-cultured with endothelial cells equally generated an angiocrine dialog, resulting in a pro-inflammatory environment, which polarized macrophages added to the co-culture towards a pro-angiogenic and pro-inflammatory state, sustaining a vicious circle [104].

Immune cells grown in 3D are also directly impacted in their gene expression, and CD4+ cells notably overexpress PD1 in 3D ECM when compared to 2D cultures [105]. Moreover, the ECM stiffness was shown to modulate the phenotype of CD4+ and CD8+ cells in the same study. In parallel, 3D collagen gels evidenced that increased collagen concentrations promoted the infiltration of regulatory T cells, in agreement with the altered ECM signaling correlated with T reg content in breast cancers [106].

Three-dimensional co-cultures also helped unravel specific interactions between cancer cells and NK cells (Figure 4B). For that purpose, breast tumor organoids derived from murine breast tumors were co-cultured with NK cells originating from the spleen [107]. Although NK cells initially repress the tumor growth and invasion, by inducing the apoptosis of invading cancer cells, NK cells exposed to tumors in vivo promoted the colony formation by tumor cells in these 3D models, which was paralleled by an increased metastatic potential in in vivo experiments. Complementarily, some models merged the microfluidic supply with an underlying 3D system to decipher the interactions between circulating neutrophils and tumor cells, as previously described in Section 2.2 [75]. Another smart system was developed, which can reversibly assemble 3D microchambers to sequentially study paracrine interactions between tumor cells and macrophages [108]. This tool notably allowed the authors to evidence that macrophages activated by distinct prostate cancer cells differentially impacted the branching of endothelial cells grown in the underlying 3D gel.

Co-culture models in matrix gels were also used to assess the efficacy of antitumoral immunotherapies. Indeed, Djikstra et al. demonstrated the feasibility of using autologous peripheral blood cells to target tumor organoids from the same patient [109]. This 3D co-culture induced tumor-specific T cell reactions that killed tumor organoids while sparing healthy organoids. The same approaches were carried out with cholangiocarcinoma organoids, with similar results [110]. Moreover, patient-derived organoids grown with an air–liquid interface, demonstrated preservation of the native infiltrated immune cells, as evidenced by the remarkable similarity in the T-cell receptor repertoire between the original tumors and their paired organoids [111]. Such tools will be instrumental in understanding which are the determinants of tumor cell sensitivity to T cells and how to improve this sensitivity in the frame of personalized medicine.

Another interesting strategy for 3D cell culture relies on the encapsulation of cells within alginate shells (Figure 4C). In that manner, cells grow as 3D clusters in which they produce their own ECM. Thanks to this kind of system, Rebelo et al. set up a co-culture of tumor cells with CAFs and monocytes [112]. In this 3D model, they recreated a pro-invasive microenvironment accumulating cytokines, matrix fibrils, and MMPs, which induced the differentiation of monocytes into M2-like activated macrophages. Finally, an acoustic-driven method was set up to generate patient-derived cell clusters made of both tumoral and immune cells, characterized by an increased secretion of IFN-γ and TNFα when compared to 2D cultures [113].

Altogether, these different 3D models should prove to be very useful to obtain a better insight into the complexity of interactions between tumors and immune cells.

### 3.3. Fueling the System, a Vital Issue

In parallel with these efforts to recreate the pro-inflammatory tumor microenvironment in 3D models, many researchers have aimed at proposing to the community perfusable models to integrate the concept of flow in 3D branched structures (Figure 4D). So far, researchers have had to use either microfluidic chips (as described in Section 2.1) or animal models to investigate this critical component in vessel physiology. In this latter case, they had to find adjustments to visualize vascular events by using translucent samples, such as a sponge in the ear of mice [114] or the zebrafish model [101]. Since institutional guidelines now strongly encourage the replacement of animal models, in agreement with the 3R rules (Replace, Refine, Reduce) of animal experimentation ethics, alternative ex vivo 3D models have been developed.

A few studies have succeeded in the proper vascularization and perfusion of organoids. In fact, vessel organoids were successfully obtained from induced Pluripotent Stem (iPS) cells, with an appropriate organization of endothelial cells, pericytes, and basement membrane, which were efficiently connected after grafting in mice and perfused [115]. Remarkably, a 3D model of neural stem cell spheroids could also be efficiently colonized by endothelial vessels, and its proper perfusion was illustrated by the diffusion of microbeads within the whole network [116]. Interestingly, the presence of vessels improved the maturation of neural stem cells and alleviated their apoptosis rate, thus reinforcing the concept of angiocriny. In the context of brain diseases, a key element of the tumor microenvironment is the BBB, which limits the access of therapies to the tumor site. Of note, in the aforementioned models, molecular markers of the BBB are expressed by the vascular structures [117], evidencing a promising potential to study how brain tumors affect the BBB, which remains a matter of debate in the community and probably depends on the brain tumor subtypes. We are convinced that such models will therefore greatly interest the neuro-oncology field.

Another approach to allow the perfusion of 3D cultures was using sacrificial carbohydrate glass to cast networks filled with endothelial cells [118]. These networks tolerated pulsatile flow and demonstrated physiological support for cells embedded in the surrounding 3D matrix, in agreement with proper tissue perfusion. In parallel, thanks to a 3D stamping method, a vascular channel was generated, which stably supported perfusion and could connect several compartments thus allowing the supply of organoids into an organ-on-a-chip system [119].

Moreover, a smart solution to generate perfusable vascular 3D models was proposed by Andrique et al. profiting from a combination of knowledge from the fields of physics and biology [120]. They used microfluidic devices to generate a triple coextrusion flow to generate so-called vesseloids, i.e., vessels covered by a soft alginate shell. Inside this shell, endothelial cells and perivascular cells self-assemble and reproduce quiescent functional vessels that can be perfused and behave as normal vessels in terms of permeability and contraction. Such a strategy offers exciting perspectives for the vascularization and perfusion of organoids. Although most of these developments were aimed at regenerative medicine, as illustrated with the graft of liver buds [121] or pancreatic islets [122], we are convinced that oncology will also benefit from the vascularization of organoids to recapitulate more relevant ecosystems. Indeed, by using “genetically reset” endothelial cells, Palikuqi et al. demonstrated that these cells could vascularize healthy or tumoral colon organoids and sustained the proliferation and arborization of these organoids [123].

Finally, to end this section on tumor vessel perfusion, it seems important to us to say a few words about vasculogenic mimicry. Based on the formation of hybrid vessels made in part by endothelial cells and in another part by tumor cells, this process was notably described in melanomas, breast cancers, and gliomas [124]. It is still a matter of debate since anatomo-pathologists hardly observe it in tumor sections. Yet, one can argue that, by its nature, this event that probably relies on progressive dedifferentiation and redifferentiation steps is difficult to detect and therefore to characterize. Interestingly, a 3D model with an endothelial channel surrounded by tumor organoids recently captured the integration of tumor cells within the pre-existing channel [125]. Such models will be helpful in shedding light on this peculiar phenomenon that may play a critical role in the interplay between the tumor and its vascular–immune microenvironment.

## 4. Discussion/Perspectives

To end this review on tumor angiogenesis and immunity models, it is time to discuss the different perspectives that the field presently offers.

From an experimental point of view, new designs are constantly generated to refine specific aspects of interest. Several limitations actually exist and restrict the potential of chips and 3D models: the choice and relevance of cells, the long-term viability of the culture, their access to high-resolution microscopy, the costs and availability of specific ECM components, etc.

Although HUVECs are still the most widespread endothelial cells used in the vessels-on-chip, iPS-derived endothelial cells are increasingly used. We hope that additional endothelial models will allow covering the variety of endothelial cell specificities associated with their tissue of origin. For instance, liver sinusoid discontinuous capillaries and impermeable brain capillaries should be made from endothelial cells with adequate permeability properties that reproduce their opposite physiology. When talking about tumor immunity, a breakthrough could also come from the integration in the experimental models of lymphatic vessels, which play a key role in immune cell homing. In that sense, lymphatic microvascular networks have been recently proposed, which could pave the way for a novel field of tumor microenvironment understanding [126]. Determining the right culture conditions, and especially the media composition allowing preservation of the differentiation state and health of many cell types together, probably represents the most significant challenge.

From a translational point of view, the proper vascularization of multicellular systems remains a bottleneck in many fields, both in oncology to mimic the physiology of tumors and in regenerative medicine to ensure the survival of tissue grafts. Both areas should therefore benefit from the efforts of both communities to improve vessel engineering and refine functional and perfusable capillaries ex vivo. Some precious knowledge may also derive from the vast number of studies on COVID-19 since endothelial activation and associated immunothrombosis have been shown to participate in the COVID-19 disease after SARS-CoV2 infection [127,128]. Incidentally, thrombosis constitutes with inflammation and angiogenesis the third partner of Virchow’s triad, which is dysregulated in different pathologies including cancers [129], and platelets are also involved in metastasis, as illustrated by the famous Trousseau syndrome. Interestingly, platelets enclose many growth factors modulating angiogenesis and immunity and can coordinate intravascular immune responses to infections and cancers [130]. The control of their degranulation is tightly orchestrated by many parameters in the organism, and is hard to reproduce in ex vivo models so far, but would represent an interesting long-term track of development.

Most models for tumor vasculature focus on neo-angiogenic mechanisms, yet alternative processes also contribute to the development of the tumoral vascular network. Indeed, tumor vessels can split after the initial formation of an endothelial pillar within the vessel lumen, in a so-called intussusception process. Interestingly, this event also seems to be associated with immune regulation since the infiltration of macrophages and T cells could be observed in the vicinity of such pillars in metastatic melanoma [131]. Moreover, tumors can also grow by hijacking pre-existing blood vessels within the surrounding tissue, and this vessel co-option was shown to be controlled by macrophage subpopulations [132]. Last, but not least, macrophages also seem to be critical for the recurrence of tumors after radiotherapy, by the recruitment of endothelial precursor cells to elicit vasculogenesis [133]. Developing new tools and models is required to elucidate how such mechanisms can modulate the vascular network and grant it a higher degree of plasticity.

Finally, immunotherapy has gained much attention over the last few years and holds great promise in oncology. For example, immune checkpoint blockade has appeared as a promising novel strategy in recent years, but its adverse effects still restrict its full clinical potential [134]. Moreover, CAR (Chimeric Antigen Receptors) T cells have revolutionized the antitumoral immunotherapy field. Yet, their immunogenicity might dampen their efficacy and/or induce anti-CAR immune responses that must be better understood to reach their full potential [135]. Much knowledge can therefore still be acquired to improve the efficiency of immunotherapy. Among the improvement tracks, the understanding of long-term maintenance of immune activity is surely a key issue, and methods to increase the longevity of NK cells in the organism are under study [136].

## 5. Conclusions

To summarize, increased knowledge of the complex interplay between tumoral, endothelial, and immune cells within the tumor microenvironment is a crucial asset for envisioning an all-out battle against tumors. Targeting multiple hits within this context could improve the overall efficacy of therapies. In that framework, the combination of VEGFR2 and PDL1 inhibitors against breast and pancreatic cancer murine models led to the generation of High Endothelial Venules and favored the infiltration and activation of lymphocytes [137]. Let us hope that this is only the first of many microenvironment-based efficient strategies in our quest toward the eradication of tumors.

## Figures and Tables

**Figure 1 cancers-14-04581-f001:**
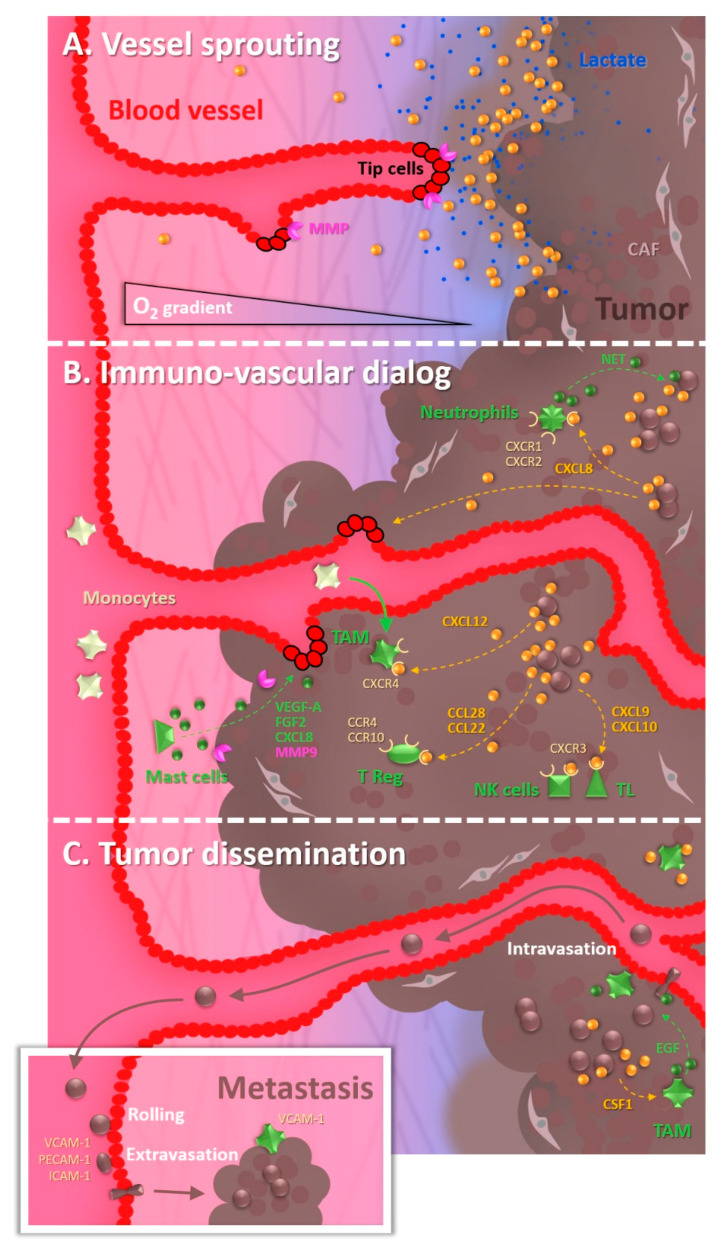
The tumor microenvironment octopus. (**A**) Tumors secrete a wide array of pro-angiogenic molecules. Hypoxia is a primary driver of angiogenesis, promoting these molecules’ expression. Moreover, lactate production by hypoxia-associated glycolysis also stimulates the invasion of endothelial tip cells. (**B**) Immune cells are recruited via the chemokines secreted by tumor cells. Although initially aimed at inhibiting tumor growth, immune cells also display several pro-tumorigenic actions. Mast cells, M2-like macrophages, and T-reg lymphocytes stimulate angiogenesis. These last two cell types also generate an immunosuppressive environment. Moreover, Neutrophil Extracellular Traps (NETs) can wrap tumor cells and form a protective shield around them. (**C**) The metastatic cascade initiates with the intravasation of tumor cells, which is promoted by the presence of macrophages. Circulating tumor cells then extravasate and colonize a distant site, in which macrophages can again help them by favoring their survival.

**Figure 2 cancers-14-04581-f002:**
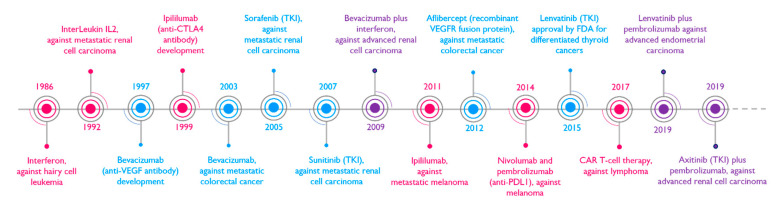
Some milestones in clinical trials against cancers based on anti-angiogenic and/or immunomodulatory strategies. This timeline integrates information about the development of anti-angiogenic (in blue) and immunotherapy (in pink) anti-cancer strategies, or the combination of both approaches (in purple), with the dates of their approval by the FDA for the treatment of specific cancers.

**Figure 3 cancers-14-04581-f003:**
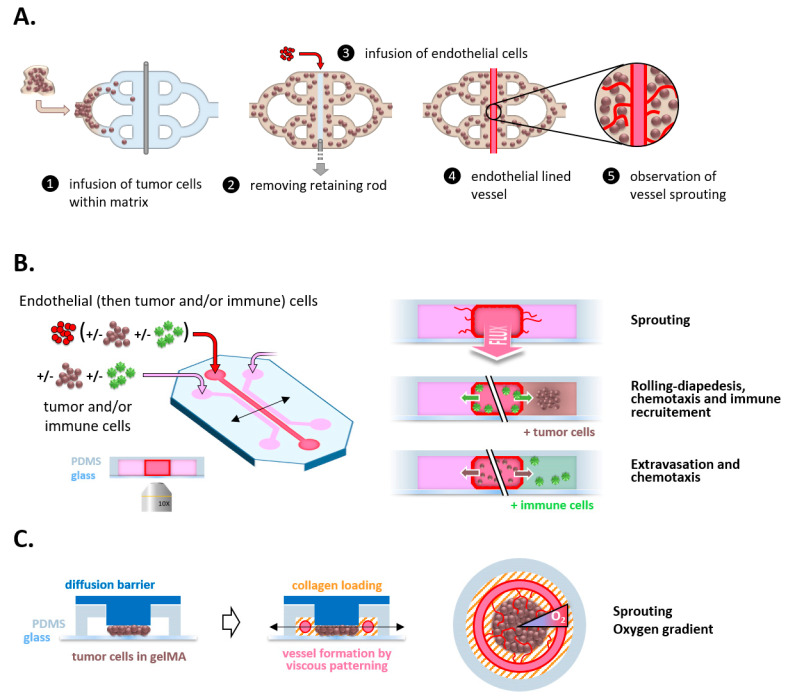
Chips shed light on the multiple facets of tumor angiogenesis and immunity. (**A**) Vascular channels can be generated by removing a needle from an ECM hydrogel. (**B**) PDMS molds classically permit generation of parallelized channels with their own reservoirs. This geometry allows investigation by microscopy of the impact upon angiogenesis of shear stress, paracrine interactions, chemotaxis, and extravasation events. (**C**) The generation of a radial oxygen gradient in a tumor disk permits deciphering of the interactions of a hypoxic tumor-on-chip with surrounding vessels and the recruitment of immune cells.

**Figure 4 cancers-14-04581-f004:**
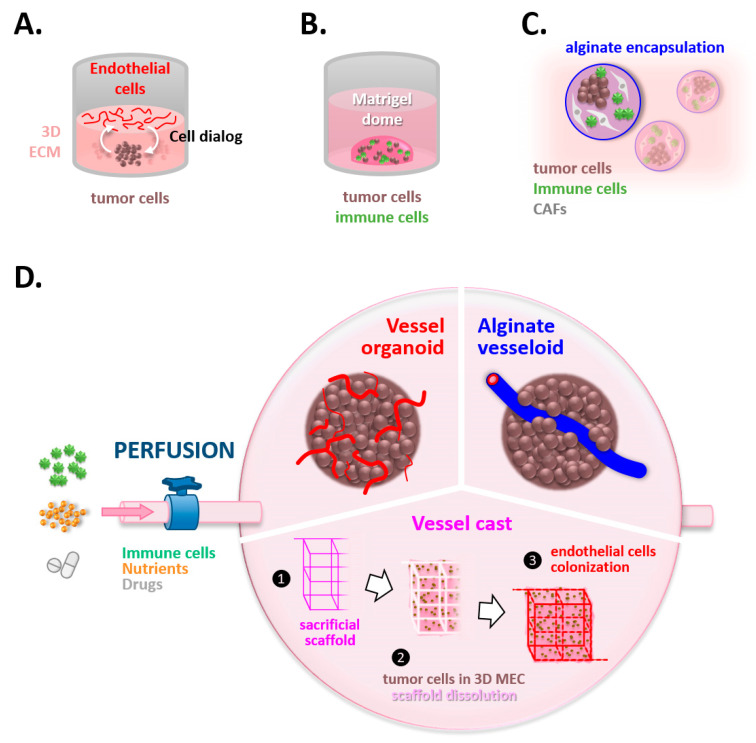
Welcome to the 3rd dimension. 3D models display a variety of opportunities to generate organized paracrine interactions. (**A**) ECM hydrogels can separate cell types to understand their paracrine interactions or migration. (**B**) Organoids preserve the tumor traits and their heterogeneity. (**C**) Alginate shells encapsulate cells and recreate a microenvironment in which cells produce their own ECM and self-organize. (**D**) In all these settings, the introduction of a vascular network represents a great challenge and allows integration of angiocrine effects and the perfusion and delivery of nutrients, cells, and drugs.

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
