# Peer review of "A Bloody Conspiracy— Blood Vessels and Immune Cells in the Tumor Microenvironment"

_cancers, 2022, doi:10.3390/cancers14194581_

Round 1

Reviewer 1 Report

The timely review by the authors has well compiled the recent findings and is appropriately written.

The authors have focused their well-designed review on the interplay of angiogenesis and immunity in cancer development. The authors have critically evaluated the role of these two key elements of the tumor microenvironment (TME). They discussed the models, namely microfluidic chips, co-culture models, and organoids, to study the immuno-angiogenic microenvironment. Finally, they also reviewed the potentiality of therapeutic targets.

Recent developments in immunotherapy have shown promise in cancer management. Understanding the mechanistic underpinnings of the TME would help develop the field.

The article is a timely review that not only compiles the role of angiogenesis in tumor immunity but also provides a critical evaluation of the recent models to study this intricate phenomenon.

Author Response

We thank the reviewer for her/his positive evaluation.

Reviewer 2 Report

The review article by TERRASSOUX et al focused on the tumor micro-environment and revealed its two features angiogenesis and immunity.

This recap is timely, as factors contributing to pro-tumorigenic activities are not well understood and the current review nicely covers the updates. Nevertheless, the comments mentioned below need to be addressed to strengthen the concept and translational view of the article.

1.     The title should change and need to make concise, to attract the attention of broad subject readers of the journal.

2.     The current review lacks information on human and clinical studies. Therefore, a separate section on human studies and past/ongoing clinical studies or trials focusing on angiogenesis with or without immunotherapy is necessary.

3.     Authors should consider adding a separate table focused on different aspects of the tumor immuno-angiogenic ecosystem that can cover the entire study comprehensively.

4.     A separate section on the limitations of the study is necessary.

5.     Several grammatical and typographical errors need to fix while re-submitting.

Author Response

We thank the reviewer for her/his positive evaluation and comments.

  1. The title should change and need to make concise, to attract the attention of broad subject readers of the journal.

The title was shortened, following the reviewer’s suggestion, to be more attractive.

  1. The current review lacks information on human and clinical studies. Therefore, a separate section on human studies and past/ongoing clinical studies or trials focusing on angiogenesis with or without immunotherapy is necessary.

We thank the reviewer for this suggestion. Although it appeared challenging to integrate a separate clinical section within this review dedicated to ex vivo models, we added a historical timeline to recapitulate milestones on this topic at the end of section 1 (lines 230-237).

  1. Authors should consider adding a separate table focused on different aspects of the tumor immuno-angiogenic ecosystem that can cover the entire study comprehensively.

This is what we integrated into Figures 1 and 2, describing these different aspects. Should we give more highlights to these terms (by increasing the font size?...)

  1. A separate section on the limitations of the study is necessary.

The limitations of the systems under the spotlight are mentioned all along the manuscript (e.g. lines 243-245, 263-265, 336-337, 433-434) and discussed with the tracks of development to overcome these issues in the perspective section. We added an explicit reference to this point in the discussion (lines 609-612).

  1. Several grammatical and typographical errors need to fix while re-submitting.The timely review by the authors has well compiled the recent findings and is appropriately written.

We apologize for these errors, and we corrected them thanks to an English language edition service, and the style was then revised by a native English speaker.

Reviewer 3 Report

This review is interesting and valuable for researchers on cancer or tissue engineering. However, despite the review, there are few sentences to be shown. Therefore, major revision should be made before re-submission. The reviewer will re-consider the manuscript only when the reviewer correctly responded to all comments.

1.

The authors should describe the components and characteristics of blood vessels, such as endothelial cells, tumor-associated endothelial cells, or pericytes.

2.

The authors should prepare the table to show the interaction and cytokines.

3.

3. Future is coming, giving a new dimension to the trade

The readers cannot understand the methodology. Why is the system important in drug research? In addition, why were the gels selected? Why were Collagen (Line 467) or alginate (line 496) appropriate for this research? The authors should describe the properties of the materials. There are some appropriate reviews to indicate the properties of the material for 3D cancer models. The authors should describe the importance of 3D models and the usage of material by quoting these related reviews.

Cancers 2020, 12 (10), 2754

http://doi.org/10.1089/ten.teb.2009.0676

4.

The authors should prepare the table for the results of the 3D models.

Author Response

Reviewer #3

This review is interesting and valuable for researchers on cancer or tissue engineering. However, despite the review, there are few sentences to be shown. Therefore, major revision should be made before re-submission. The reviewer will re-consider the manuscript only when the reviewer correctly responded to all comments.

We thank the reviewer for her/his constructive comments.

  1. The authors should describe the components and characteristics of blood vessels, such as endothelial cells, tumor-associated endothelial cells, or pericytes.

We modified the text accordingly to define these features of blood vessels and their alterations in tumors (lines 52-58).

  1. The authors should prepare the table to show the interaction and cytokines.

We thank the reviewer for this suggestion. We integrated this table (between lines 163 and 164) to give an overview of this feature.

  1. 3. Future is coming, giving a new dimension to the trade

The readers cannot understand the methodology. Why is the system important in drug research? In addition, why were the gels selected? Why were Collagen (Line 467) or alginate (line 496) appropriate for this research? The authors should describe the properties of the materials. There are some appropriate reviews to indicate the properties of the material for 3D cancer models. The authors should describe the importance of 3D models and the usage of material by quoting these related reviews.

 Cancers 2020, 12 (10), 2754

 http://doi.org/10.1089/ten.teb.2009.0676

We thank the reviewer for this suggestion. We integrated a description of some biophysical properties of the materials and added new references, including the suggested smart reviews in our manuscript (lines 448-467).

  1. The authors should prepare the table for the results of the 3D models.

We, unfortunately, did not understand the purpose of this comment. This review explains the interest of 3D models to answer questions about tumor angiogenesis and immunity, but there are few studies, investigating specific questions, and such disparate “results” are difficult to gather in an overview table.

Round 2

Reviewer 2 Report

The authors addressed all comments satisfactorily.

Reviewer 3 Report

Nice revise.